

**Global assessment of how averaging over land-surface heterogeneity affects modeled evapotranspiration rates**
Elham Rouholahnejad Freund[1,2], Ying Fan[3], James W. Kirchner[2,4,5]
[1]Laboratory of Hydrology and Water Management, Ghent University, Ghent, Belgium
[2]Department of Environmental Systems Science, ETH Zurich, 8092, Zurich, Switzerland
[3]Department of Earth and Planetary Sciences, Rutgers University, New Brunswick, NJ, United States
[4]Swiss Federal Research Institute WSL, Birmensdorf, 8903, Switzerland
[5]Dept. of Earth and Planetary Science, University of California, Berkeley, CA 94720, United States
*Correspondence to*: Elham Rouholahnejad Freund, elham.rouholahnejad@gmail.com
**Key points**
- Evapotranspiration rates and the properties that regulate them are spatially heterogeneous at scales
orders of magnitude smaller than typical Earth System Models (ESMs) grid cells. Averaging over this spatial
heterogeneity may lead to biased estimates of energy and water fluxes in ESMs.
- We quantified the effects of averaging over spatial heterogeneity on grid-cell-averaged evapotranspiration
(ET) rates over heterogeneous landscapes across the globe and highlighted the locations where the
heterogeneity bias matters. We showed that because the relationships driving ET are nonlinear, averaging
over sub-grid heterogeneity of derivers of ET, namely precipitation (P) and potential evapotranspiration
(PET), leads to overestimation of average ET.
- Our analysis showed that this "heterogeneity bias" is most pronounced in mountainous terrain, in
landscapes where P is inversely correlated with PET, and in regions with temperate climates and dry
summers.
- We showed that the magnitude of this heterogeneity bias grows on average, and expands over larger
areas, as the size of the grid cell increases.



**Abstract**
The major goal of large-scale Earth System Models (ESMs) is to understand and predict global change. However,
computational constraints require ESMs to operate on relatively large spatial grids (typically ~1 degree or ~100 km
in size) with the result that the heterogeneity in land surface properties and processes at smaller spatial scales
cannot be explicitly represented. Averaging over this spatial heterogeneity may lead to biased estimates of energy
and water fluxes in ESMs. For example, evapotranspiration rates and the properties that regulate them are spatially
heterogeneous at scales orders of magnitude smaller than typical ESM grid cells. Here we quantify the effects of
spatial heterogeneity on grid-cell-averaged evapotranspiration (ET) rates, as seen from the atmosphere over
heterogeneous landscapes across the globe. In an earlier study, we used a Budyko framework to functionally relate
ET to precipitation (P) and potential evapotranspiration (PET), and used a sub-grid closure relation to quantify the
effects of sub-grid heterogeneity on average ET at 1° by 1° grid cells- the scale of typical ESM. We showed that
because the relationships driving ET are nonlinear, averaging over sub-grid heterogeneity in P and PET leads to
overestimation of average ET. In this study, we extend that work to the globe and examine the global distribution of
this bias, its scale dependence, and the underlying mechanisms. Our analysis shows that this "heterogeneity bias" is
more pronounced in mountainous terrain, in landscapes where P is inversely correlated with PET, and in regions
with temperate climates and dry summers. We also show that the magnitude of this heterogeneity bias grows on
average, and expands over larger areas, as the size of the grid cell increases. Correcting for this overestimation of
ET in ESMs is important for modeling the water cycle, as well as for future temperature predictions, since current
overestimations of ET rates imply smaller sensible heat fluxes, and potential underestimation of dry and warm
conditions in the context of climate change. Our work provides a basis for translating the heterogeneity bias into
correction factors in large-scale ESMs, and highlights the regions where more detailed mechanistic modeling is
needed.




## 1. Introduction

Earth System Models (ESMs) are designed to understand interactions between the land surface, atmosphere, and oceans and to predict global environmental changes. However, the Earth system and its underlying physical processes are highly heterogeneous across orders of magnitude in scale below the scale of typical ESM grids (e.g., 1° by 1°). Despite increasing recognition of the need to mechanistically represent physical processes in ESMs, currently even the most disaggregated large-scale ESMs cannot explicitly represent the spatial heterogeneity of land surface hydrological properties at scales that are important to atmospheric fluxes. Overlooking this spatial heterogeneity and instead averaging over land surface properties at the scale of ESM model grid cells may have important implications for water and energy flux estimates in large-scale ESMs (Avissar and Pielke, 1989; Giorgi and Avissar, 1997; Ershadi et al., 2013; Lu et al., 2014).

Estimates of evapotranspiration (ET) fluxes have significant implications for future temperature predictions. Smaller ET fluxes imply greater sensible heat fluxes and therefore, amplified dry and warm conditions in the context of climate change (Seneviratne et al., 2010). Surface evaporative fluxes (and thus energy partitioning over land surfaces) are nonlinear functions of available water and energy, and thus are coupled to spatially heterogeneous surface characteristics (e.g., soil type, vegetation, topography) and meteorological inputs (e.g., radiative flux, wind, and precipitation) (Kalma et al., 2008; Shahraeeni and Or, 2010; Holland et al., 2013). These characteristics are spatially variable on length scales of <1 m to many kilometers, well below typical ESM grid scales of ~100 km. ESMs calculate grid-averaged surface and atmospheric fluxes from grid-averaged land surface parameterizations (Sato et al., 1989; Koster et al., 2006; Santanello and Peters-Lidard, 2011). Thus ET estimates that are derived from spatially-averaged land surface properties do not capture ET variations driven by the underlying surface heterogeneity (McCabe and Wood, 2006). Because the relationships driving ET are nonlinear, the average ET flux from a heterogeneous landscape may be different from an ET estimate calculated from spatially averaged inputs (Rouholahnejad Freund and Kirchner, 2017).

Several studies have quantified the effects of land surface heterogeneity on ET, potential evapotranspiration (PET), and latent heat (LH) fluxes, and have found that averaging over land surface heterogeneity can potentially bias ET estimates either positively or negatively. For example, Boone and Wetzel (1998) studied the effects of soil texture variability within each pixel in the Land-Atmosphere-Cloud Exchange (PLACE) model, which has a spatial resolution of approximately 100 by 100 km. They reported that accounting for sub-grid variability in soil texture reduced global ET by 17%, increased total runoff by 48%, and increased soil wetness by 19%, compared to using a homogenous soil texture to describe the entire grid cell. Kollet (2009) found that heterogeneity in soil hydraulic conductivity had a strong influence on evapotranspiration during the dry months of the year, but not during months with sufficient moisture availability. Hong et al. (2009) reported that aggregating radiance data from 30 m to 60, 120, 250, 500, and 1000 m resolution (input upscaling) and then calculating ET from these aggregated inputs at these grid scales using Surface Energy Balance Algorithm for Land (SEBAL, Bastiaanssen et al., 1998a) yields



slightly larger ET estimates as compared to ET calculated with finer resolution inputs and then aggregated at the
desired grid scales (output upscaling). The discrepancy between ET estimated with the output upscaling method
and the input upscaling method grows as the size of the grid-cell increases (the difference between ET calculated
from the input and output upscaling methods is ~20% more at a grid scale of 1 km by 1 km compared to grid scale
of 120 m by 120 m). Aminzadeh et al. (2017) investigated the effects of averaging surface heterogeneity and soil
moisture availability on potential evaporation from a heterogeneous land surface including bare soil and vegetation
patches. They found that if the heterogeneity length scale is smaller than the convective atmospheric boundary
layer (ABL) thickness, averaging over heterogeneous land surfaces has only a small effect on average potential
evaporation rates. Averaging over larger-scale heterogeneities, however, led to overestimates of potential
evaporation.

McCabe and Wood (2006) found that remote sensing retrievals of ET are larger than the corresponding in-situ flux
estimates and characterized the roles of land surface heterogeneity and remote sensing resolution in the retrieval
of evaporative flux. McCabe and Wood (2006) used Landsat (60 m), Advanced Space borne Thermal Emission and
Reflection Radiometer (ASTER) (90 m), and MODIS (1020 m) independently to estimate ET over the Walnut Creek
watershed in Iowa. They compared these remote sensing estimates to eddy covariance flux measurements and
reported that Landsat and ASTER ET estimates had a higher degree of consistency with one another and correlated
better to the ground measurements (0.87 and 0.81, respectively) than MODIS- based ET estimates did. All three
remote sensing products overestimated ET as compared to ground measurements (at 12 out of 14 tower sites).
Upon aggregation of Landsat and ASTER retrievals to MODIS scale (1 km), the correlation with the ground
measurements decreased to 0.75 and 0.63 for Landsat and ASTER, respectively.

Contrary to overestimation bias, many remotely sensed ET estimates that include parameters related to
aerodynamic resistance are significantly affected by heterogeneity, and underestimate ET as the scale increases
(Ershadi et al., 2013). Because aerodynamic resistance is significantly affected by land surface properties (e.g.,
vegetation height, roughness length, and displacement height), decreases in aerodynamic resistance at coarser
resolutions could lead to smaller estimates of evapotranspiration. Ershadi et al. (2013) showed that input
aggregation from 120m to 960 m in Surface Energy Balance System (SEBS, Su, 2002) leads to up to 15 %
underestimation of ET at the aggregated grid resolution in an study area in the south-east of Australia.
Rouholahnejad Freund and Kirchner (2017) quantified the impact of sub-grid heterogeneity on grid-average ET
using a simple Budyko curve (Turc, 1954; Mezentsev, 1955) in which long-term average ET is a non-linear function
of long-term averages of precipitation (P) and potential evaporation (PET). They showed mathematically that
averaging over spatially heterogeneous P and PET results in overestimation of ET (Fig. 1). Their analysis implies that
large-scale ESMs that overlook land surface heterogeneity will yield biased evapotranspiration estimates due to the
inherent nonlinearity in ET processes. They did not, however, estimate the likely magnitude of this heterogeneity
bias beyond a few example grid cells.






The recognition that spatial averaging can potentially lead to biased flux estimates has prompted methods for
representing sub-grid-scale heterogeneities and processes within ESMs. Accounting for land surface heterogeneity
in large-scale ESMs is constrained by limitations in both computational power (Baker et al. 2017) and the availability
of high-resolution forcing data. There have been several attempts to integrate sub-grid heterogeneity in ESMs while
maintaining the computational costs affordable. In "mosaic" approaches, the model is run separately for each
surface type in a grid cell, and then the surface specific fluxes are area-weighted to calculate the grid-cell average
fluxes (e.g., Avissar and Pielke, 1989; Koster and Suarez, 1992). The "effective parameter" approach (e.g., Wood
and Mason, 1991; Mahrt et al., 1992), by contrast, seeks to estimate effective parameter values at the grid cell
scale that subsume the effects of sub-grid heterogeneity. Estimating these effective parameters can be challenging
because the relevant land-surface processes typically depend nonlinearly on multiple interacting parameters, and
land-surface signals at different scales are propagated and diffused differently in the atmosphere. Alternatively, the
"correction factor" approach (e.g., Maayar and Chen, 2006) uses sub-grid information on spatially heterogeneous
land-surface processes and properties to estimate multiplicative correction factors for fluxes that are originally
calculated from spatially averaged inputs at the grid-cell scale. All three approaches try to reduce the
heterogeneous problem to a homogeneous one that has equivalent effects on the atmosphere at the grid-cell
scale.

There is a growing need to understand how sub-grid heterogeneity and the atmosphere's integration of it, affect
grid-scale water and energy fluxes, and to develop effective methods to incorporate these effects in ESMs (Clark et
al., 2015, Fan et al., 2019). The above-mentioned studies present the potential effects of spatial heterogeneity on
water and energy flux estimates in land surface models at several scales, but are deficient in proposing a general
framework for quantifying systematic biases in ET estimates due to averaging over heterogeneities. In a previous
study, we used the Budyko framework as a simple estimator of ET, and demonstrated theoretically how averaging
over heterogeneous precipitation and potential evapotranspiration at the grid scale of a typical ESM (e.g., 1° by 1°)
can lead to systematic overestimation of long-term average ET fluxes from heterogeneous landscapes. In the
present study, we apply that analysis across the globe and highlight the locations where the heterogeneity bias
matters. Our hypotheses are that, (1) strongly heterogeneous landscapes, such as mountainous terrain, will exhibit
higher bias due to averaging, (2) the bias will be higher in climates where P and PET are inversely correlated in
space, and (3) heterogeneity bias will decrease as the spatial scales of averaging decrease.

**2. Effects of sub-grid heterogeneity on ET estimates in the Budyko framework**
Budyko (1974) showed that the long-term annual average evapotranspiration is a function of both the supply of
water (precipitation, P) and the evaporative demand (potential evapotranspiration, PET) under steady-state
conditions and in catchments with negligible changes in storage (Eq. 1; Turc, 1954; Mezentsev, 1955).





$$ET = f(P, PET) = \frac{P}{\left( (\frac{P}{\overline{PET}})^n + 1 \right)^{1/n}}. \quad (1)$$

Evapotranspiration rates are inherently bounded by energy and water limits. Under arid conditions ET is limited by
the available supply of water (the water limit line in Fig. 1b), while under humid conditions ET is limited by
atmospheric demand (PET) and converges toward PET (the energy limit line in Fig. 1b). Budyko showed that over a
long period and under steady-state conditions, hydrological systems function close to their energy or water limits.
These intrinsic water and energy constraints make the Budyko curve downward-curving.

In a heterogeneous landscape, like the simple example of two ESM columns in Fig. 1a, P and PET vary spatially. The
two columns with heterogeneous P and PET are represented by the two solid black circles on the Budyko curve in
Fig. 1b. In this hypothetical two-column example, the true average of ET values calculated from individual
heterogeneous inputs (the solid black circles) lies below the curve (the grey circle, labeled "true average").
However, if we aggregate the two columns and consider the system as one column with average properties, the
function of average inputs (averaged P and PET over the two columns) lies on the Budyko curve (the open circle)
which is larger than the true average of the two columns. In short, in any downward curving function, the function
of the average inputs (the open circle) will always be larger than the average of the individual function values (the
true average; grey circle). The difference between the two can be termed the "heterogeneity bias".
Rouholahnejad Freund and Kirchner (2017) showed that when nonlinear underlying relationships are used to
predict average behaviour from averaged properties, the magnitude of the resulting heterogeneity bias can be
estimated from the degree of the curvature in the underlying function and the range spanned by the individual data
being averaged. The second-order, second-moment Taylor expansion of the ET function f(P,PET) (Eq. 1) around its
mean directly yields:
$$\bar{f}(P, PET) = \overline{ET} \approx f(\bar{P}, \overline{PET}) + \frac{1}{2} \frac{\partial^2 f}{\partial P^2} var(P) + \frac{1}{2} \frac{\partial^2 f}{\partial PET^2} var(PET) + \frac{\partial^2 f}{\partial P\, \partial PET} cov(P, PET) \quad , \quad (2)$$
where $\bar{f}(P, PET)$ is the true average of the spatially heterogeneous ET function, $f(\bar{P}, \overline{PET})$ is the ET function
evaluated at is average inputs $\bar{P}$ and $\overline{PET}$ , and where the derivatives are quantified at $\bar{P}$ and $\overline{PET}$. Evaluating the
derivatives using Eq. (1) and reshuffling the terms, Rouholahnejad Freund and Kirchner (2017) obtained the
following expression for the heterogeneity bias, the difference between the average ET, $\bar{f}(P, PET)$, and the ET
function evaluated at the mean of its inputs, $f(\bar{P}, \overline{PET})$:
$$f(\bar{P}, \overline{PET}) - \bar{f}(P, PET) \approx (n + 1) \frac{\bar{P}^{n+1} \overline{PET}^{n+1}}{(\bar{P}^n + \overline{PET}^n)^{2+1/n}} \left[ \frac{1}{2} \frac{var(P)}{\bar{P}^2} + \frac{1}{2} \frac{var(PET)}{\overline{PET}^2} - \frac{cov(P, PET)}{\bar{P}\,\overline{PET}} \right]. \quad (3)$$
As shown by Fig. 1b and Eq. (2), the discrepancy between average of the ET function and the ET function of the
average inputs (the heterogeneity bias) is proportional to both the degree of nonlinearity in the function, as
defined by its second derivatives, and the range of variation in its input variables, as defined by their variances. Eq.
(3) allows one to estimate how much the curvature of a nonlinear relationship and the variance of its inputs at any



desired scale will affect estimates of the true mean. However, to the best of our knowledge, the consequences of
these nonlinearities for global evaporative flux estimates have not previously been quantified.

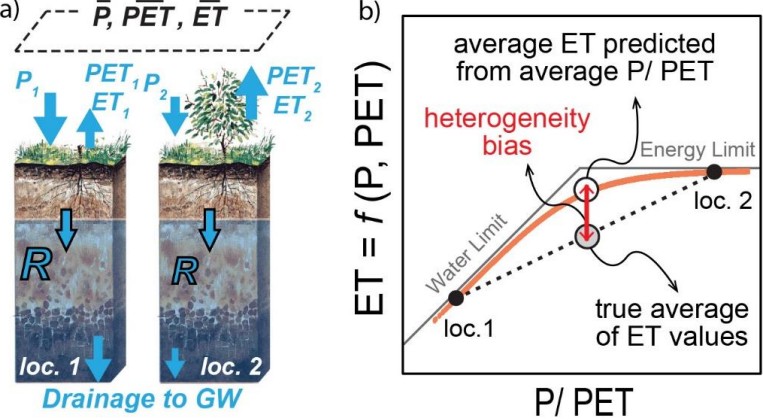


Figure 1. Heterogeneity bias in a hypothetical two-column model in the Budyko framework. The true average ET of
the columns (gray circle) lies below the curve and is less than the average ET estimated from the average P/PET of
the two columns (open circle). The heterogeneity bias depends on the curvature of the function and the spread of
its inputs.

**3. Effects of sub-grid heterogeneity on ET estimates at 1° by 1° grid scale across the globe**
Across a landscape of size similar to a typical ESM grid cell (1° by 1°), soil moisture, atmospheric demand (PET) and
precipitation (P) will vary with topographic position; hillslopes will typically be drier, and riparian regions will be
wetter. To quantify the likely biases introduced by averaging over this land surface heterogeneity, we used the
approach outlined in section 2 to the global land surface area at 1° by 1° grid scale. Within each 1° by 1° grid cell,
we used 30 arc-second values of P (WorldClim; Hijmans et al., 2005) and PET (WorldClim; Hijmans et al., 2005) to
examine the variations in small-scale climatic drivers of ET. Because 30 arc-seconds is nearly 1 km, hereafter we
refer to 30 arc-second data as 1km values for simplicity. The spatial distribution of long-term annual averages
(1960-1990) of P and PET values at 1 km resolution and 1km values of the aridity index (AI=P/PET) are shown in Fig
2a-c. ET estimated from these 1km P and PET values using Eq. 1 are then averaged at 1° by 1° scale ("true average",
Fig. 2e). To mimic the averaging that takes place within ESMs, we also averaged the 1km values of P and PET within
each grid cell and then modeled ET using Eq. 1 applied to these averaged input values. The difference between
these two ET estimates is the heterogeneity bias.

We also calculated the heterogeneity bias using Eq. (3), which describes how the nonlinearity in the governing
equation and the heterogeneity in P and PET jointly contribute to the heterogeneity bias. The heterogeneity bias





estimates obtained by direct calculation and by Eq. (3) were functionally equivalent ($R^2$=0.97, root mean square
error of 0.17%).

Fig. 3a-d illustrates the variability (quantified by standard deviation) of 1km values of P, PET, aridity index, and
altitude at the 1° by 1° grid scale. The heterogeneity bias in long-term average ET fluxes at the 1° by 1° grid scale
(Fig. 3e) highlights regions around the globe where ET fluxes are likely to be systematically overestimated. The
spatial distribution of the heterogeneity bias (Fig. 3e) closely coincides with locations with large variability in the
aridity index (Fig. 3c), which is driven in turn by topographic variability (Fig. 3d). Strongly heterogeneous landscapes
exhibit significant heterogeneity biases in long-term average ET fluxes, although the global average heterogeneity
bias is small (<1%). Physically based ET calculations may exhibit larger heterogeneity biases than the modest values
we calculate here, because the Budyko approach already subsumes spatial heterogeneity effects at the catchment
scale (and also temporal heterogeneity effects due to its steady- state assumptions). The heterogeneity bias in ET
estimates shown in Fig. 3e corresponds to long-term average ET estimates. Given the fact that P and PET can vary
temporally (i.e., seasonality), the estimated bias could be much larger, particularly where P and PET are inversely
correlated (see the last term of Eq. 3).

Our results show that the topographic gradient and hence the variability in aridity index across a desired grid size
exhibit consistent, predictable patterns of associated prediction bias in evapotranspiration estimates at that scale.


Figure 2. Global distribution of one-kilometer resolution annual mean precipitation (a: P; WorldClim; Hijmans et al.,
2005), potential evapotranspiration (b: PET; WorldClim; Hijmans et al., 2005), aridity index (c: AI=P/PET; WorldClim;
Hijmans et al., 2005), topography (d: SRTM; Jarvis et al., 2008), and (e) evapotranspiration (ET) at 1° by 1° scale by
averaging 1km values of ET calculated using the Budyko function (Eq. 1).

241





Figure 3. Global spatial distribution of variability (standard deviation) of one-kilometer values of a) precipitation (P), b) potential evapotranspiration (PET), c) aridity index (AI=P/PET), and d) altitude at 1° by 1° grid cell. The approximated averaging bias in ET estimates (e) is calculated using Eq. (3). Grid cells with large standard deviation in altitude and aridity index encounter higher percentage of averaging bias.





**4. Variation in heterogeneity bias across climate zones, data sources, and grid scales**

With increased availability of spatial data, it is becoming standard practice to assess input data uncertainties and their propagated impacts on water and energy flux estimates in land surface models. To quantify how choices among alternative input data products could affect the heterogeneity bias in ET estimates, we calculated the heterogeneity bias at 1 ° by 1° grid cell resolution across the contiguous US using four different pairs of P and PET data products. Two precipitation data sets, Prism (http://prism.oregonstate.edu) and WorldClim (Hijmans et al., 2005), along with two PET data sets, MODIS (Mu et al., 2007) and WorldClim (Hijmans et al., 2005), all at 1 km resolution, were combined in all possible pairs. The heterogeneity bias in ET estimates (Eq. 3), as outlined in section 2, was evaluated from 1km values of P, PET, and the estimated average ET using the Budyko relationship (Eq. 1) for each of the four input data pairs. Fig. 4a-e compares the spatial distributions of heterogeneity bias across the contiguous US for the four pairs of P and PET data products. The heterogeneity bias in ET estimates reached as high as 36 % in the western US using Prism P and WorldClim PET as input to the ET model (Fig. 4a). A visual comparison of Figs. 4a, c, d, and e shows that the choice of P data source (Prism vs. WorldClim) had a bigger effect on the heterogeneity bias than the choice of PET data source (MODIS vs. WorldClim). In all cases, data sources that were more variable in relation to their means (Prism for P and WorldClim for PET; Fig. 4b) led to larger heterogeneity biases, as expected from Eq. (3). If we had conducted our global analysis (Fig. 3) with Prism P and either WorldClim or MODIS PET we would have obtained larger heterogeneity biases, but Prism P is not freely available globally.

If we divide the heterogeneity biases shown in Fig. 4 by Köppen-Geiger climate zones (Peel et al., 2007; Fig. 5), we see that the heterogeneity bias is distinctly higher in particular climate-terrain combinations. The heterogeneity bias is higher in regions with temperate climate and dry summers (climate zone Cs) and in regions with cold, dry summers (climate zone Ds) perhaps due to the sharp spatial gradient in their water and energy sources for evapotranspiration. These areas typically have high topographic relief, combined with seasonal climate. The heterogeneity effects on ET estimates in these regions are expected to be even higher when a mechanistic model of ET is used. We expect that averaging over temporal variations of drivers of ET, especially in places with strong seasonality, could bias the ET estimates but can not be quantified in the Budyko framework due to its underlying steady-state assumptions. Figure 5 also illustrates the relative magnitudes of the heterogeneity biases obtained with the four pairs of P and PET data sources. The heterogeneity bias generally decreases in the order: Prism P-WorldClim PET >> Prism P-MODIS PET >> WorldClim P-WorldClim PET >> WorldClim P-MODIS PET.




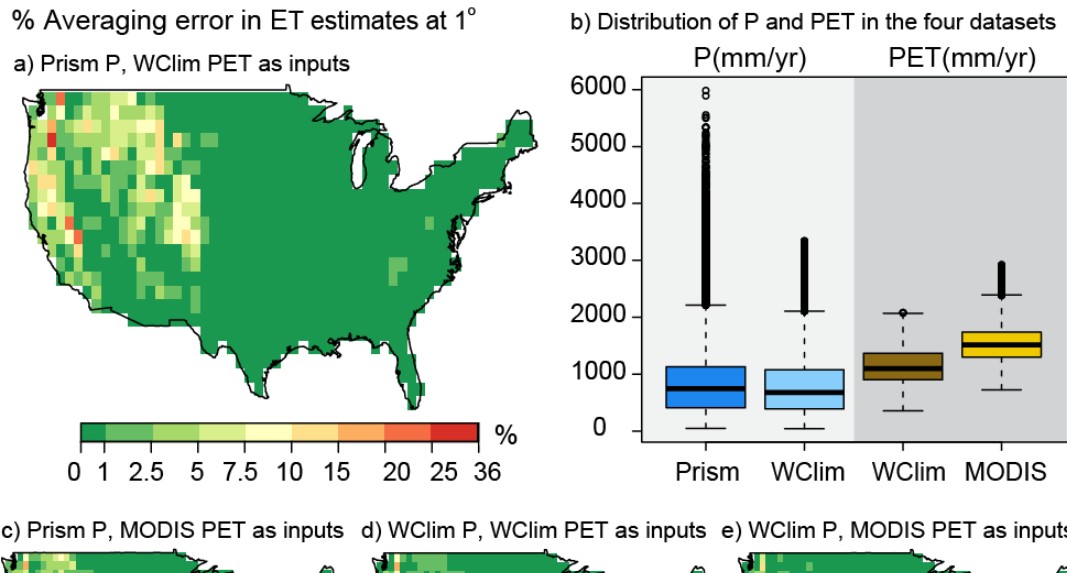


Figure 4. Estimated averaging bias (Eq. 3) across contiguous US using one-kilometer values of a) Prism P and
WorldClim PET c) Prism P and MODIS PET d) WorldClim P and WorldClim PET, and e) WorldClim P and MODIS PET as
inputs. The distribution of P and PET in the four datasets is shown in b).






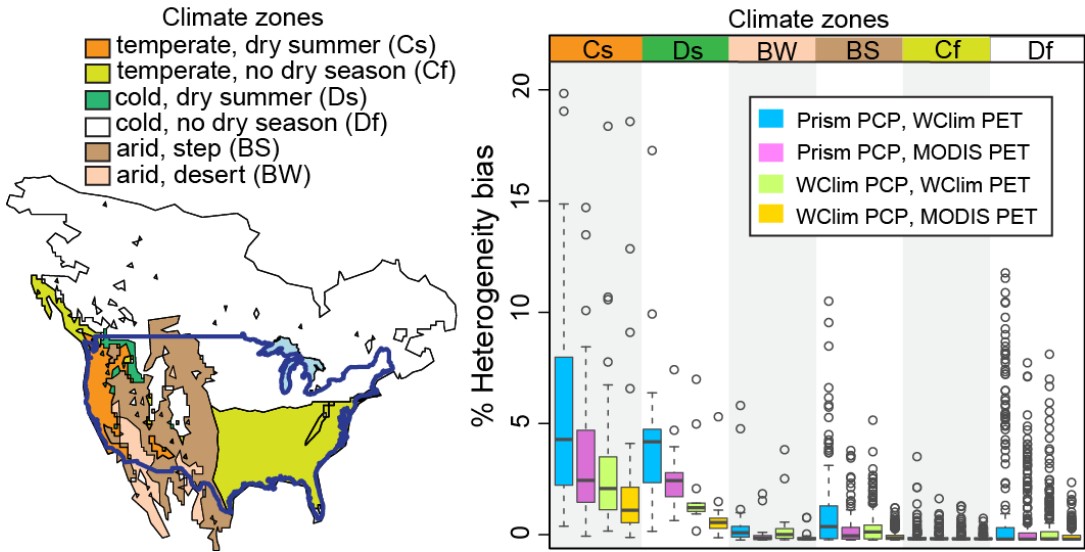


Figure 5. Köppen-Geiger climate classification (Peel et al., 2007 in Beck et al. 2013) across contiguous US and the
distribution of corresponding calculated averaging bias in ET estimates (Eq. 3) at 1° by 1° grid cell at individual
climate zone shown by boxplot. The background color code in the box plot corresponds to the climate zones on the
left.  Three data points with heterogeneity biases of over 20% are off-scale.

One may expect that future increases in computing power will lead to ESMs with smaller grid cells than those in
common usage today.  It is therefore useful to ask how changes in ESM grid resolution are likely to affect the
heterogeneity biases that we have estimated in this paper.  To quantify the heterogeneity bias in ET estimates as a
function of grid scale, we repeated our analysis at various grid resolutions using Switzerland as a test case.  We
started with high-resolution (500m) maps of long-term average annual precipitation and PET across the Swiss
landscape (Fig. 6), and then used Eq. 3 to estimate the heterogeneity bias at grid scales ranging from 1/32° to 2° (~3
km to ~200 km).  As Fig. 6 shows, aggregating P and PET over larger scales leads to larger, and more widespread,
overestimates in ET.  Conversely, at finer grid resolutions, the average heterogeneity bias is smaller, and the
locations with large biases are more localized.




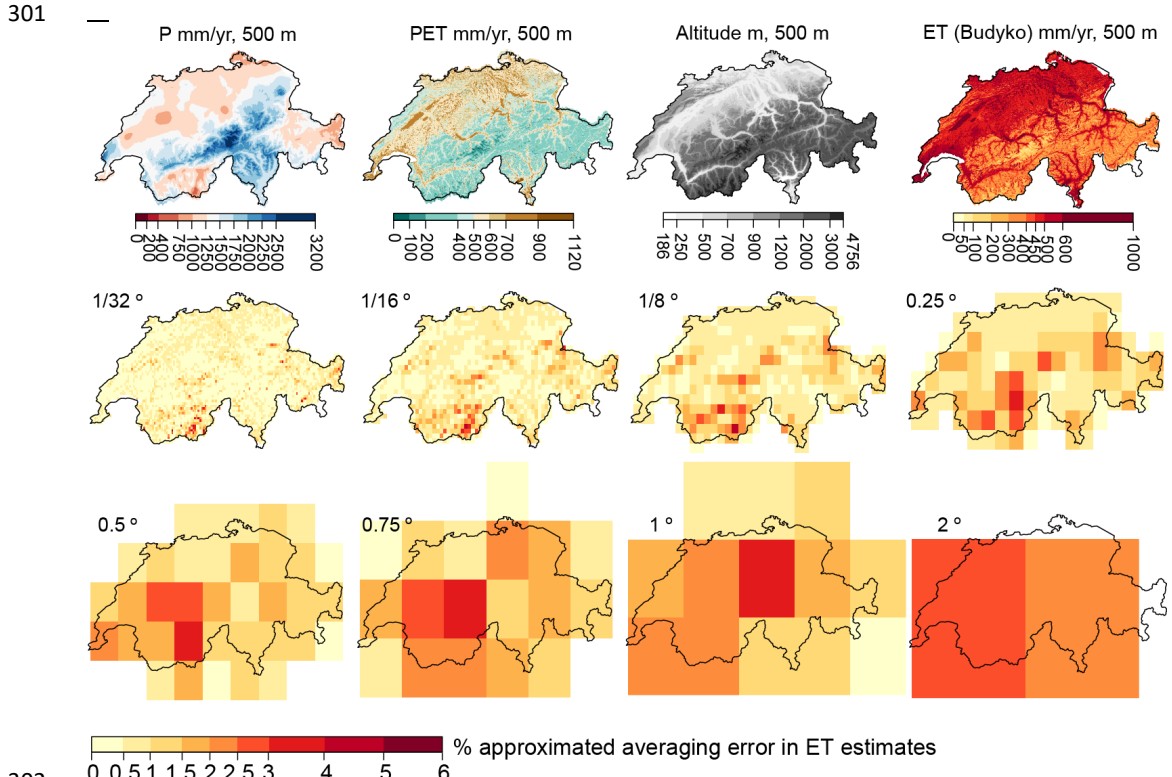


Figure 6. Heterogeneity bias in ET estimates at various scales across Switzerland, estimated from 500m climate
data. ET is calculated using the Budyko relationship (Eq. 1). Heterogeneity bias was estimated from 500m
precipitation (P) and potential evapotranspiration (PET), and their variances at each grid scale, using Eq. 3. At finer
grid resolutions, the heterogeneity bias is more localized, and smaller on average.






**5. Summary and discussion**
Because evapotranspiration (ET) processes are inherently bounded by water and energy constraints, over the long
term, ET is always a nonlinear function of available water and PET, whether this function is expressed as a Budyko
curve or another ET model. These nonlinearities imply that spatial heterogeneity will not simply average out in
predictions of land surface water and energy fluxes in ESMs. Overlooking the spatial heterogeneity in large scale
ESMs could lead to biases in estimated water and energy fluxes (e.g. ET rates). Here we have shown that, across
several scales, averaging over spatially heterogeneous land surface properties and processes leads to biases in
evapotranspiration estimates. These biases can be estimated, and these estimates can potentially be used as
correction factors to improve calculations of surface-atmosphere water and energy fluxes across landscapes.

In this study, we used Budyko curves as simple models of ET, in which long-term average ET rates are functionally
related to long-term averages of P and PET. We used an approach outlined by Rouholahnejad Freund and Kirchner
(2017) to estimate the heterogeneity bias in modeled ET at 1-degree grid scale across the globe (Fig. 3), and also at
multiple grid scales across Switzerland (Fig. 6), using finer-resolution P and PET values as drivers of ET. We showed
how the heterogeneity effects on ET estimates vary with the nonlinearity in the governing equations and with the
variability in land surface properties. Our analysis shows that heterogeneity effects on ET fluxes matter the most in
areas with sharp gradients in the aridity index, which are in turn controlled by topographic gradients, and not
merely in areas that are either arid or humid (e.g., compare Fig. 3e with Fig. 2c).

According to our analysis, regions within the U.S. that have temperate climates and dry summers exhibit greater
heterogeneity bias in ET estimates (Fig. 5). We show that the heterogeneity bias in ET estimates at each grid scale
depends on the variance in the drivers of ET at that scale (Fig. 4), and on the choice of data sources used to
estimate ET. Heterogeneity bias was significantly larger across the contiguous United States when P and PET data
sources with larger variances were used (Fig. 4).

We also explored the magnitude and spatial distribution of heterogeneity bias in ET estimates as a function of the
scale at which the climatic drivers of ET are averaged. We found that as heterogeneous climatic variables are
aggregated to larger scales, the heterogeneity biases in ET estimates become greater on average, and extend over
larger areas (Fig. 6). At smaller grid scales, the heterogeneity bias does not completely disappear, but instead
becomes more localized around areas with sharp topographic gradients. Finding an effective scale at which one can
average over the heterogeneity of land surface properties and processes has been a longstanding problem in Earth
science. Our analysis shows that at smaller resolutions the average heterogeneity bias as seen from the
atmosphere becomes smaller, but there is no characteristic scale at which it vanishes entirely (Fig. 6). The
magnitude and spatial distribution of this bias depend strongly on the scale of the averaging and degree of the
nonlinearity in the underlying processes. The averaging bias concept is general and extendable to any convex or





concave function (Rouholahnejad Freund and Kirchner 2017), meaning that in any nonlinear process, averaging
over spatial and temporal heterogeneity can potentially lead to bias.

One should keep in mind that the true mechanistic equations that determine point-scale ET as a function of point-
scale water availability and PET (if such data were available) may be much more nonlinear than Budyko's empirical
curves, because these curves already average over the spatial heterogeneities across spatial and temporal scales.
Thus, we expect that the real-world effects of sub-grid heterogeneity are probably larger than those we have
estimated in Sects. 3 and 4 of this study. In addition, the 1km P and PET values that are used in our global analysis
might be still too coarse to represent small-scale heterogeneity that is important to evapotranspiration processes.

Budyko curves are empirical relationships that functionally relate evaporation processes to the supply of water and
energy under steady-state conditions in closed catchments with no changes in storage. Our analysis likewise
assumes no changes in storage, nor any lateral transfer between the model grid cells, although both lateral
transfers and changes in storage may be important, both in the real world and in models. Unlike the Budyko
framework, ET fluxes in most ESMs are often physically based (not merely functions of P and PET) and are
calculated at much smaller time steps (seconds to minutes). These models often represent more processes that are
important to evapotranspiration (such as storage variations and lateral transfers) and include their dynamics to the
extent that is computationally feasible. Because these relationships may be much more nonlinear than Budyko
curves, there may also be significant averaging biases when complex physically based models are used to estimate
ET from spatially aggregated data. Therefore, we are now working to quantify aggregation bias in ET fluxes using a
more mechanistic land surface model.

Our results have further implications for representing sub-grid heterogeneity in hydrological parameterizations of
large scale ESMs, for example as sets of correction factors. However, the estimated bias shown in this study is for
long-term average ET estimates using a conceptual model that uses long-term annual averages. Average ET could
be substantially affected by temporal heterogeneity in water and energy fluxes, particularly in climates with strong
seasonally and shifts between water-limited and energy-limited conditions. The temporal variations in the drivers
of ET fluxes have not been addressed in the current study but can potentially be a source of bias for ET flux
estimates. Estimating aggregation bias in ET fluxes at time scales that are relevant to ESMs is therefore needed.
Once such bias estimations are quantified at daily or sub-daily time scales, they can be used as correction factors to
account for the aggregation bias in ET flux estimates.

**Acknowledgements**
E.R.F. acknowledges support from the Swiss National Science Foundation (SNSF) under Grant No. P2EZP2_162279.
The authors thank Massimiliano Zappa of the Swiss Federal Research Institute WSL for providing the 500m
resolution data that enabled the analysis shown in Fig. 6.





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
