# Peer review of "Global assessment of how averaging over spatial heterogeneity in precipitation and potential evapotranspiration"

_Hydrology and Earth System Sciences, 2019_

## Referee Comment (RC1) · Anonymous Referee #1 · 16 Apr 2019

The manuscript presents a global-scale assessment of the effects (i.e., bias) of large-scale averaging of atmospheric forcing, namely P and PET, on evapotranspiration rates. The assessment relies on the application of a previously published approach based on the Budyko framework and using existing datasets of P and PET. Additional analyses are presented at the continental- (i.e., CONUS) and regional-scale (i.e., Switzerland) in order to explain the role of climate and the scaling of evapotranspiration bias with the grid resolution, respectively.

The work addresses an important subject of research (i.e., how to quantify averaging effects in large-scale models), which is certainly of relevance for HESS readership and

for a broad scientific community. However, in its current form the work appears too much a mere application without offering new insights on such a relevant subject. In my opinion, this makes the contribution (after substantial revisions) more suitable for a technical note. I provide below a list of points that support my evaluation.

1. Authors motivate the need of their work (i.e., quantification of averaging effects on evapotranspiration) and discuss their results in light of well-known ESMs simplifications that do not take into account the fine-scale spatial heterogeneity in the atmospheric forcing and land surface characteristics. However, the averaging effects are assessed under steady-state conditions and neglecting non-linear land surface processes, two assumptions that do not reflect the actual way ESMs (and large-scale integrated models) are implemented. While authors clearly disclose these limitations (see lines 225-226; lines 229-230; lines 271-274 lines 361-364, etc...), my impression is that at the end of the article the reader is left to wonder about the maturity of the work. For instance it is not clear how current results can be exploited to calculate bias correction factors for ESMs simulations. Sentences as those reported at lines 363-364 ("... we are now working to quantify aggregation bias in ET...) support somehow my impression that reported results are in a sort of "intermediate" stage.

2. In a similar vein to the previous point, one of the main conclusions is that from 'an atmospheric perspective' averaging is for sure an approximation in those regions characterized by strong topographic gradients. This is probably not new and authors could have made an effort to explain better the different degree of sensitivity between P and PET.

3. Another point of the work (summarized in Figure 4) is that using different datasets we end up with different averaging effects. Again, was this not expected? Could authors provide an explanation of the different degree of sensitivity between P and PET datasets? If you do not provide any insight on this how can you claim (in the abstract) that your work discusses the underlying mechanisms of such differences? I have the same concern for Figure 5. The ordering (see lines 275-276) is really not informative.

4. According to Figure 5 it seems that when estimating ET the spatial heterogeneity matters for a minor portion of CONUS domain and when using certain datasets. Is this realistic? These results seem in conflict with efforts (of several groups) that have emphasized the need of including lateral moisture distribution in ESM simulations. Including this lateral effects may completely change the "picture", am I wrong? This concern brings me to the introduction (lines 145-146) when authors highlight the need of a general framework for systematically quantifying biases in ET estimates due to spatial averaging. Is it the case at the current stage of the work?

5. The discussion around Figure 6 is not clear. If I compare at single grid points the color scale at 1/32°, 1/16°, 1/8° and so on with the ones at 1° and 2°, I see that the magnitude of the bias is not increasing, isn't? Further, the bias extends over larger areas because you're increasing the resolution, am I wrong? In any case, here it is important to implement some statistics that accurately quantify the scaling of the bias with the grid resolution.

Specific points: - The title is a bit misleading because you're accounting just for land surface heterogeneity related to P and PET.

- Key points cannot be defined as long and multiple sentences

- In the abstract you cannot make a long discussion of a previous work findings. Please revise.

- Line 70: What do you mean with "grid-averaged land surface parameterizations"?

- Paragraph between lines 99-108 appears disconnected from the main flow of the introduction.

- Lines 126-128: The issue of spatial averaging is also due to the fact that atmospheric and land surface components of ESMs "work" at different resolutions. In order words, it is not " just" a problem of data volume and computational limitations.

- Line 122: what do you mean with "likely" magnitude?

- In Eq. 1 "n" is not defined. How is it estimated?

- How PET is calculated in the two datasets? This point has to be discussed in order to provide additional information about possible discrepancy between the many existing approaches to calculate PET

- Figure 3e and Figure 4: How did you calculate the percentage averaging error?

- Lines 269-270: Saying that "perhaps due to the sharp gradient. . ." is a weak statement that conveys the impression that you're not really sure about the interpretation of the results.

---

## Author Comment (AC1) · 3 May 2019

**Hydrol. Earth Syst. Sci.**
**hess-2019-103**

**Dear Reviewer #1,**
**Thank you for your review and the detailed comments. Following please find our point by point response to your suggestions and questions. The Reviewer's comments are in regular font and our response is in bold.**

**Response to Referee #1**
The manuscript presents a global-scale assessment of the effects (i.e., bias) of large-scale averaging of atmospheric forcing, namely P and PET, on evapotranspiration rates. The assessment relies on the application of a previously published approach based on the Budyko framework and using existing datasets of P and PET. Additional analyses are presented at the continental- (i.e., CONUS) and regional-scale (i.e., Switzerland) in order to explain the role of climate and the scaling of evapotranspiration bias with the grid resolution, respectively.

The work addresses an important subject of research (i.e., how to quantify averaging effects in large-scale models), which is certainly of relevance for HESS readership and for a broad scientific community. However, in its current form the work appears too much a mere application without offering new insights on such a relevant subject. In my opinion, this makes the contribution (after substantial revisions) more suitable for a technical note. I provide below a list of points that support my evaluation.

> **We thank the reviewer for his/her interest in this work. We leave it up to the Editor to determine whether it would be more appropriate for this work to appear as a technical note or a research article.**

1. Authors motivate the need of their work (i.e., quantification of averaging effects on evapotranspiration) and discuss their results in light of well-known ESMs simplifications that do not take into account the fine-scale spatial heterogeneity in the atmospheric forcing and land surface characteristics. However, the averaging effects are assessed under steady-state conditions and neglecting non-linear land surface processes, two assumptions that do not reflect the actual way ESMs (and large-scale integrated models) are implemented. While authors clearly disclose these limitations (see lines 225-226; lines 229-230; lines 271-274 lines 361-364, etc: : :), my impression is that at the end of the article the reader is left to wonder about the maturity of the work. For instance it is not clear how current results can be exploited to calculate bias correction factors for ESMs simulations. Sentences as those reported at lines 363-364 (": : : we are now working to quantify aggregation bias in ET: : :) support somehow my impression that reported results are in a sort of "intermediate" stage.

> **Our research in this area is indeed continuing, in ways that build on the present paper. We nonetheless think it is useful because it demonstrates, at global scale, an overall framework for estimating how averaging over heterogeneity in atmospheric forcing at the land surface affects evapotranspiration estimates. We use Budyko framework even though it is not actually used in ESM's, because it is a simple "see through" test case for quantifying these biases. These results are clearly not directly transferrable as estimates of the averaging biases in ESM's – which, indeed, we suspect are substantially larger, for reasons that we explain in the paper.**

> **We state in the paper that the current results can not be directly exploited by ESMs to correct for averaging bias, although the proposed methodology sheds light on the potential**

**ways one can account for this bias (depending on what ET calculation algorithm ESMs or land surface models use and the scales they average over sub-grid heterogeneities) (line 374-382). We will try to make the revised paper even more explicit on this point.**

**We agree that it would be more informative to estimate the averaging bias for physically based, time-varying ET calculations as used in ESM's. Our paper highlights a general methodology that can be used to estimate the systematic bias due to averaging, but not the precise magnitude of this bias because it will change significantly depending on ET formulation used.**

2. In a similar vein to the previous point, one of the main conclusions is that from 'an atmospheric perspective' averaging is for sure an approximation in those regions characterized by strong topographic gradients. This is probably not new and authors could have made an effort to explain better the different degree of sensitivity between P and PET.

**Equations 7 and 8 of Rouholahnejad-Freund and Kirchner (2017) show that averaging biases in Budyko ET estimates are equally sensitive to the same percentage variability in P and PET. Thus we do not try to explain the different degree of sensitivity, per se, between P and PET (because, at least in percentage terms, these sensitivities are not different in the Budyko approach). If P is more variable than PET (in percentage terms) across landscapes, then the variability in P will make a larger contribution to the averaging bias. (At least in the Budyko approach; whether this is true for more physically based ET estimates remains to be seen.)**

3. Another point of the work (summarized in Figure 4) is that using different datasets we end up with different averaging effects. Again, was this not expected? Could authors provide an explanation of the different degree of sensitivity between P and PET datasets? If you do not provide any insight on this how can you claim (in the abstract) that your work discusses the underlying mechanisms of such differences? I have the same concern for Figure 5. The ordering (see lines 275-276) is really not informative.

**Briefly, in figure 4 the choice of P datasets is more consequential than the choice of PET datasets, because the two P datasets differ more from one another (specifically, the percentage variability in P is substantially bigger in PRISM than in WorldClim), whereas the two PET datasets are about equally variable. If desired, we can include a discussion of these points in the revised manuscript (although this will probably require also revisiting the mathematical development of the earlier paper, and thus would add somewhat to the length).**

4. According to Figure 5 it seems that when estimating ET the spatial heterogeneity matters for a minor portion of CONUS domain and when using certain datasets. Is this realistic? These results seem in conflict with efforts (of several groups) that have emphasized the need of including lateral moisture distribution in ESM simulations. Including this lateral effects may completely change the "picture", am I wrong? This concern brings me to the introduction (lines 145-146) when authors highlight the need of a general framework for systematically quantifying biases in ET estimates due to spatial averaging. Is it the case at the current stage of the work?

**We treated lateral transfers in some detail in our 2017 paper and we currently have nothing to add on this subject. Lateral transfers may of course be important, but unless and until we have reliable quantitative estimates of how big lateral transfer fluxes actually are (and where they are), it will be difficult to estimate their impact on ET heterogeneity biases.**

5. The discussion around Figure 6 is not clear. If I compare at single grid points the color scale at 1/32, 1/16, 1/8 and so on with the ones at 1 and 2, I see that the magnitude of the bias is not increasing, isn't? Further, the bias extends over larger areas because you're increasing the resolution, am I wrong? In any case, here it is important to implement some statistics that accurately quantify the scaling of the bias with the grid resolution.

**These inferences are correct, and are stated in the text and the figure caption (although in different language). We will try to make the paper even more explicit on these points. We can add a panel to this figure that plots the mean error as a function of the grid resolution.**

Specific points: - The title is a bit misleading because you're accounting just for land surface heterogeneity related to P and PET.

**Potentially can change to: Global assessment of how averaging over heterogeneity in precipitation and potential evapotranspiration affects modeled evapotranspiration rates**

- Key points cannot be defined as long and multiple sentences
**Key points will be shortened in the revised manuscript.**

- In the abstract you cannot make a long discussion of a previous work findings. Please revise.
**We will streamline the abstract in the revised manuscript.**

- Line 70: What do you mean with "grid-averaged land surface parameterizations"?
**To our understanding, the authors used parameter values that correspond to the average of the given property over each grid cell.**

- Paragraph between lines 99-108 appears disconnected from the main flow of the introduction.
**The paragraph is on the application of remotely sensed data in estimating ET at several scales. We will try to make the connection clearer.**

- Lines 126-128: The issue of spatial averaging is also due to the fact that atmospheric and land surface components of ESMs "work" at different resolutions. In order words, it is not " just" a problem of data volume and computational limitations.
**We can bring this point into the introduction of the revised manuscript.**

- Line 122: what do you mean with "likely" magnitude?
**This will be revised to "potential".**

- In Eq. 1 "n" is not defined. How is it estimated?
**"n" is a dimensionless parameter that modifies the partitioning of P between E and Q and was assumed to have a value of n=2 in our analysis (literature value). This will be added to the manuscript.**

- How PET is calculated in the two datasets? This point has to be discussed in order to provide additional information about possible discrepancy between the many existing approaches to calculate PET

        **The Worldclim PET dataset (Hijmans et al., 2005) is based on the Hargreaves method (Hargreaves and Allen 2003). The MODIS PET product (Mu et al, 2007) is based on Penman–Monteith equation. The text will be revised to include this information.**

- Figure 3e and Figure 4: How did you calculate the percentage averaging error?
        **Figure 3e is the approximated averaging error calculated using Eq. 3. This is written in the figure caption and will be added to the main text too. The estimates of biases obtained by direct calculation of average ET from finer resolution data were functionally equivalent to those obtained by Eq. 3 (R2=0.97), as we state on lines 214-217**

- Lines 269-270: Saying that "perhaps due to the sharp gradient: : :" is a weak statement that conveys the impression that you're not really sure about the interpretation of the results.
        **We will revise this.**

---

## Referee Comment (RC2) · Anonymous Referee #2 · 3 Jun 2019

This is a nice little study with predictable results following equation 1 to 3. Following Rouholahnejad Freund and Kirchner (2017), this paper was expected. One could write many more papers like that using different functions for ET. It is well written and easy to follow. As the authors note in the conclusions, the actual interesting aspect is using this approach to correct ET estimates. The rest of the conclusions are straightforward. Since this paper needed to be written, without contributing really new insights, I suggest to publish it following the suggestions of the other reviewer. Note, Figure 1b is almost identical to figures 3 and 6 in Rouholahnejad Freund and Kirchner (2017). This has not been indicated and is not good practice.

---

## Author Comment (AC2) · 17 Jun 2019

**Hydrol. Earth Syst. Sci.**
**hess-2019-103**

**Response to Referee #2**

**Dear Reviewer #2,**
**Thank you for your comments on the manuscript. Following please find our response to your suggestion. The Reviewer's comments are in regular font and our response is in bold.**

**Response to Referee #2**
This is a nice little study with predictable results following equation 1 to 3. Following Rouholahnejad Freund and Kirchner (2017), this paper was expected. One could write many more papers like that using different functions for ET. It is well written and easy to follow. As the authors note in the conclusions, the actual interesting aspect is using this approach to correct ET estimates. The rest of the conclusions are straightforward. Since this paper needed to be written, without contributing really new insights, I suggest to publish it following the suggestions of the other reviewer. Note, Figure 1b is almost identical to figures 3 and 6 in Rouholahnejad Freund and Kirchner (2017). This has not been indicated and is not good practice.

**We thank the reviewer for his/her interest in this work. We will add "adapted from Rouholahnejad-Freund and Kirchner, 2016" to the figure caption of Fig.1.**

---

## Author Response (AR1)

**Hydrol. Earth Syst. Sci.**
**hess-2019-103**

**Response to editor's comments**

**Dear Editor,**

**Thank you very much for your detailed comments. Following please find our point-by-point response to your questions and suggestions. The editor's comments are in regular font and our response is in bold. The page and line numbers refer to the revised manuscript that will be submitted with this response (with "all mark up" display for review).**

The authors present a study where they investigated the effects of spatial averaging on modelled evaporation (ET) estimates at the global scale. They make use of the Budyko framework to model ET and use this same framework to spatially 'average' ET and to determine the heterogeneity bias. This method was already presented in a previous paper (Rouholahnejad Freund & Kirchner, 2017), but is now applied at the global scale. Only applying a method to the global scale without adding new insights (or very limited) is not enough for a new publication. As also mentioned by reviewer #1, this should be improved. Clearly show the added value of this study above the previous one.

- **Our previous work showed mathematically that averaging over spatially heterogeneous P and PET results in overestimation of ET within the Budyko framework. We did not, however, determine where around the globe, and under what conditions, this heterogeneity bias is likely to be most important. In this work, we examine the global distribution of this bias, its scale dependence, and its sensitivity to variations in P versus PET.**

- **Our goal is to identify where, under what conditions, and at what spatial scales averaging over heterogeneities in P and PET could be most important to estimates of evapotranspiration, but not to quantify the absolute magnitude of these averaging effects.**

- **Our work outlines a strategy for quantifying heterogeneity biases and potentially correcting for them, and highlights regions where more detailed mechanistic modeling is needed.**

- **Our analysis of percent variability of P and PET products shows that percent variabilities of precipitation products are in general larger than PET products and hence contribute more to heterogeneity bias.**

- **Our analyses show that mountainous terrain, regions with temperate climates and dry summers, and landscapes where spatial variations in precipitation and potential evapotranspiration are inversely correlated exhibit greater heterogeneity bias in ET estimates.**

- **Our analysis of scale dependence shows that heterogeneity bias increases almost exponentially as gird cell sizes increase.**

- **The second order Taylor expansion of a function around its mean is a powerful quantifiable approach that can be practically used in estimating biases in ET calculations due to spatial averaging over heterogeneous inputs. We use Budyko relations as ET functions for purposes of demonstration, but discuss that the approach is expandable and quantifiable in any other ET function at large scales (global, continental, and regional scales). We used Budyko as an**

example "see-through" case to show the applicability of the proposed mathematical method at scales that are relevant to large-scale land surface models.

- One can use this approach to correct for averaging bias without explicitly representing finer-scale processes within the modeling framework. The same approach can potentially be used in more mechanistic ET models with time varying inputs at each modeling time step (daily or sub-daily).

**We have revised the manuscript to point out the added value of this paper more clearly.**

In the abstract (L40-41) it is promised to proved insights in the underlying mechanisms, but these can not (or limited) be found in the manuscript.

**We revised the manuscript in way that it doesn't emphasize the underlying mechanisms (because of the inherent characteristics of Budyko Framework used in this paper). The discussion about the sensitivity of heterogeneity bias to climatic variability (variability in P and PET) is now added to the revised manuscript ( P14, L326-339). We now no longer mention underlying mechanisms.**

Furthermore, I have some doubts on your methodology. You frame your study in such way, that it will help to quantify errors in ET due to spatial averaging for ESMs. To investigate this, you don't use a ESM, but you choose for the Budyko framework for simplicity reasons. However, the Budyko framework is a first order estimate, meant for large catchments under steady-state (see also comment reviewer #1). I wonder if the gridcells can be considered as mini-catchments under steady state.

**In response to reviewer #1's comments, we made it clearer in the manuscript that the current heterogeneity bias rates are not applicable to correct for this bias in ESMs because ESMs use different algorithms to calculate ET at daily or sub-daily temporal resolution, which goes beyond the steady-state assumptions of Budyko curves. We nonetheless think the current manuscript is useful because it demonstrates, at global scale, an overall framework for estimating how averaging over heterogeneity in atmospheric forcing at the land surface affects evapotranspiration estimates. We state in the paper that the current results can not be directly exploited by ESMs to correct for averaging bias, although the proposed methodology sheds light on the potential ways one can account for this bias (depending on the specific ET algorithms ESMs use and the scales at which they average over sub-grid heterogeneities) (P21, L394-402).  We made the revised paper even more explicit on this point. Our paper highlights a general methodology that can be used to estimate the systematic bias due to averaging, but not the precise magnitude of this bias because it will change significantly depending on ET formulation used.**

**Regarding Budyko's steady sate assumptions: here we used long-term averages of P and PET to get long-term averages of ET at grid scales. Over long periods of time, changes in storage are commonly neglected in water balance calculations. Besides, current large-scale physically**

based models overlook changes in deep groundwater storage and lateral transfers of water among their vertical columns at any given modeling time step, so they force grid cells to behave like catchments, whether they do so in reality or not.

We agree that on shorter time steps, Budyko curves cannot be used over individual grid cells. The temporal variations in climatic variables and the effect of their averaging on ET estimates is indeed an open question but cannot be addressed within Budyko framework.

We revised the manuscript to include these points more clearly ( P21, L394-402 and P22, line440-450).

What is the effect of lateral flow, irrigation, advected energy (dry gridcell next to wet gridcell) etc.? This should be more discussed in detail.

We treated lateral transfers in some detail in our 2017 paper and we currently have nothing to add on this subject. Lateral transfers may of course be important, but unless and until we have reliable quantitative estimates of how big lateral transfer fluxes actually are (and where they are), it will be difficult to estimate their impact on ET heterogeneity biases. These shortcomings are stated in the discussion part of the manuscript (P22, L440-443).

Furthermore, I wonder if the study shows the real ET-heterogeneity bias (as far as you can at all), or that I am looking at uncertainties in rainfall products? Because looking at figure 1b, the bias is largest once P/PET deviates most between 2 locations. This is the case when either P or PET differs most between locations. Often PET differs less than P, so the bias is dominated by differences in P (as shown in figure 4). Hence I am not surprised to see that the bias is related to topography, because it is well known that P changes significantly with altitude (and also becomes more uncertain).

Equations 4 of the revised manuscript shows that averaging biases in Budyko ET estimates are equally sensitive to the same underline{percentage} variability in P and PET. Thus we do not try to explain the different degree of sensitivity, per se, between P and PET (because, at least in percentage terms, these sensitivities are the same). If P is more variable than PET (in percentage terms) across landscapes, as is often the case, then the variability in P will make a larger contribution to the averaging bias. (At least in the Budyko approach; whether this is true for more physically based ET estimates remains to be seen, and we are working on this question.)

As a practical matter, it is difficult to know whether rainfall products overestimate the spatial variability in P (due to errors or uncertainties), or underestimate it (by leaving out mechanisms that cause real-world variability in P). We agree that it is an interesting question (how variability in real-world P relates to variability in rainfall product P), but that is well beyond the scope of this paper.

Figure 4a is the boxplot of two products of P and PET for the entire US. It is the spatial (and temporal) variability in each calculation unit (grid cells) that contributes to the heterogeneity bias in ET estimate on that grid cell. The purpose of the figure was to show the difference in mean and variability of the data products. We have revised the manuscript to more clearly

**explain that the key variables are the fractional variability in P and PET, and that the fractional variability in P will usually be the dominant variable (P14, L326-339).**

In relation to this, I wonder if figure 5b would not look similar once you plot climate zone against the standard deviation in P?

**The derived heterogeneity bias term (Eq. 3) is a direct function of percentage standard deviations in P and PET (see Eq. 4). Standard deviations of P and PET at 1-degree grid scale show similar patterns (figure 3 a and b). We expanded Fig. 5 to show how the distributions of P and PET change as a function of climate zones, both in absolute terms (Figs. 5c and 5d) and in percentage terms (Fig. 5e). This was one of the concerns of the first reviewer too. We made changes in the manuscript to make sure these points are stated more clearly (P14, L326-339 and Figure 5c-e).**

Based on these concerns, plus the 2 critical recommendations by the reviewers I advise to do a major revision of your manuscript and emphasize what we can learn from this study in addition to the previous study (what are the new insights). Hereafter, I will send out the manuscript for a new review round. Please also have a careful look at the comments of reviewer #1. They will help to improve the quality of the paper.

**The added values of this work are reviewed in page one and two of the current document in response to the Editor's first comment.**

Specific comments:
- P1L23: what do you mean by "landscapes where P is inversely related with PET"?

**landscape in which spatial variations in P are inversely related to spatial variations in PET. This is corrected in the revised manuscript.**

- P3L78: in my view ET and LH are synomyms.

**We corrected this in the revised manuscript.**

- P5L151-153: the hypotheses miss link with previous text. Please make the connection clearer.

**The strongest link is actually with the analysis that follows in Section 2, and we have added that linkage explicitly to this sentence (P6, L172-176).**

- fig 1b: Please change this figure to the common way Budyko curves are drawn, i.e.: aridity on x-asis (aridity=PET/P) and not the wetness index

**Budyko curves are drawn both ways (either PET/P and ET/P on the x and y axes, or as we have done it here and in our previous paper, P/PET and ET/PET on the x and y axes). We agree with you that the official UN/FAO definition of P/PET as the "aridity index" is a source of confusion, but that train left the station years ago and none of us can solve it now. We clearly refer to**

the "aridity index" and not "aridity".  While in theory one could plot the curves the other way, that would create a lot of confusion with respect to our original paper – among other things, it would require entirely different equations to connect the heterogeneity bias to the Budyko plot, and readers would not understand why the equations are different between the two papers.  So we really think it is essential to keep the axes the way they are.  Besides, for the same reason you pointed out above – that P is more variable than PET – we think it is much more intuitive to have the main "driving variable", P, on only one axis rather than redundantly on both the x and y axes.

- P7L208: P/PET is not aridity, become once P/PET becomes larger the index becomes wetter instead of dryer

We did not say "aridity", we said "aridity index", and the distinction is important.  As mentioned above, AI=P/PET became a widely used international definition of the "aridity index" (not aridity) years ago, and that can't be reversed now.  (We agree that it is confusing to have an "aridity index" that is really a wetness index, but that's not a problem we created and for the reasons explained above, P/PET is really the correct x-axis variable for our problem.  Given that the figures and text are written based on this definition (AI=P/PET) and the index has been defined several times throughout the text, we would like to keep the definition as it is.

- P11L262-264: this line somehow suggest that you prefer prism P, because you then have larger biases? Why?

Our intention in this section is not to suggest any particular data product merely because it gives a larger bias.  Our point was that we can't show explicitly that Prism P gives a larger heterogeneity bias at global scale, because the data set is not available globally without paying a substantial fee.  We see that the phrasing was unclear and we have clarified it.

- Fig 4: I would swop a) and b)

Figure 4 is revised as suggested.

- Fig 5b: would be nice to also make this graph for standard deviation of P and PET. Likely it's similar.

Three panels are added to figure 5 as requested.

- P15L314: add comma after e.g.

Revised as suggested.

[revised manuscript text omitted]

---

## Author Response (AR2)

**Hydrol. Earth Syst. Sci.**
**hess-2019-103**

**Response to editor's comments – report #2**

**Dear Editor,**

**Thank you very much for your comments. Following please find our point-by-point response to your questions and suggestions. The editor's comments are in regular font and our response is in bold. The page and line numbers refer to the revised manuscript that will be submitted with this response (with "all mark up" display for review).**

As can be seen by the two review reports, both reviewers still have major concerns. Especially, the concerns about the parameter choice n are important. Since you study scale dependency, having a constant n is dangerous. Please provide information on:
1) the value of n
2) was n constant?
3) how did you determine it?
4) was it spatially variable?
5) how sensitive is the approach to n?

**This is an important point and we should have mentioned the reason to hold n constant.**

**We did not vary n either in space or in time, because doing so would create artifacts that would confound the effects of spatial heterogeneity in P and PET. For example, if we vary n from place to place, then how do we separate the effects of spatial heterogeneity from the effects of the imposed variation in n? For similar reasons, we do not agree that having a constant n is "dangerous" for a study of scale-dependency. Instead, it is essential for n to be held constant because otherwise one cannot separate the effects of scale-dependent variation in P and PET from the effects of scale-dependent variation in n. Recall that in our analysis we use Budyko curves as an analytical framework (or a simple "see-through" function) for exploring the consequences of spatial heterogeneity in landscape properties. We do not quantify the heterogeneity bias in ESMs (which are not based on Budyko curves), nor do we use Budyko curves as a proxy for what ESM ET estimates would be.**

**In the revised version we present a sensitivity analysis (for n values ranging from 2 to 5) in the supplement (Figs S1, S2) and discuss its main results in the manuscript (under Summary and Discussion). Those results are: 1) the spatial patterns of aggregation bias are similar, 2) the absolute magnitude of aggregation bias increases somewhat for higher values of n, as predicted by Eqs. 3 and 4, and 3) the Taylor approximation in Eqs. 3 and 4 yield realistic estimates of the aggregation bias for all values of n that were tested.**

Furthermore, spatial heterogeneity in P and PET should be better explained as indicated in report #2, reviewer #1 (first point). As well as the temporal dynamics (report #1, reviewer#3).

**Please see our response to reviewer #1 first point.**

Lastly, I am doubting whether this manuscript should be transferred into a technical note as suggested by reviewer #1. In principle I agree, since the manuscript presents a method to define scaling issues and the 'research component' is a bit on the back. I leave it up to the authors to decide whether they prefer a technical note or a research article. However, if you chose the latter, please emphasis the research component (what can we learn from it, e.g., process understanding).

We still think that it is appropriate for this manuscript to be published as a research article. We explain the added value of the manuscript in response to the second last comment by reviewre#1, report#2.

Hydrol. Earth Syst. Sci.
hess-2019-103

Response to Referee #1, report#2

**We thank Reviewer #1 for her/his comments on the manuscript, and present our responses below.   The Reviewer's comments are in regular font and our responses are in bold.**

Response to Referee #1
I now read the revised manuscript version and rebuttal letter of the manuscript entitled "Global assessment of how averaging over spatial heterogeneity in precipitation and potential evapotranspiration affects modeled evapotranspiration rates". I appreciate authors effort in improving the manuscript and clarifying several aspects of their work. Having said that, I must also admit that I am still not fully convinced by some of the arguments the authors used in the reply letter to present (and defend) their work. Therefore, I highlight below those important points that need to be addressed:

- Authors treat P and PET as model input data defined at the same scale (see lines 292-294). However, this does not reflect how typically ESMs dynamical cores are designed and/or have been evolving. Forcing terms (e.g., precipitation, temperature, humidity) are defined at the atmospheric model grid (usually coarser) while PET is calculated at the PFT-level using land surface features (e.g., LAI, aerodynamic resistance). In light of this, variability in modelled P and PET occurs at different spatial scales. This makes, in my opinion, the calculation of a correction factor for ET less straightforward. Could authors elaborate on this point?

**The introduction describes "mosaic" approaches in which PET (and ET) are calculated for individual PFTs and then aggregated. In any case, our purpose is not to mimic the way that ESMs actually calculate ET (obviously so, since ESMs do not employ Budyko curves).  Our purpose is instead to illustrate how variability in P and PET would be translated into biased ET estimates, using Budyko curves as a simple "see-through" function for illustration purposes.  In the case that the reviewer mentions (P and PET calculated at different scales), the magnitude of the bias would depend on whether the PET estimates were first averaged at the atmospheric grid cell scale (Case 1), or whether P and PET were jointly used to estimate ET for each PFT within each grid cell, and then these ET estimates were averaged (Case 2).  The aggregation bias would be greater in Case 1 than in Case 2, but we don't think we would be justified in going into these details in the present paper, because, again, the Budyko calculations presented here are in any case not the way that ESMs actually calculate ET. We make sure we emphasize this point in the manuscript.**

- Authors affirm that their work highlights under which climate conditions averaging in P and PET has an effect on ET estimates. Without giving clear explanations I suspect that the largest differences found for Cs and Ds climate zones are still mainly driven by topography. Note also that this analysis is limited to the CONUS domain where there are not many sampling points for certain climate zones (e.g., Ds). This information should be provided in the plots and the statistical significance of the differences should be tested. An analysis at the global scale would be certainly more convincing.

**We agree that the large aggregation biases found in the Cs and Ds climate zones are mainly driven by topography  We highlight this in the second paragraph of Section 4: "Heterogeneity biases are higher in regions with temperate climates and dry summers (climate zone Cs) and in regions with cold, dry summers (climate zone Ds), most likely due to the sharp spatial gradient in their water**

and energy sources for evapotranspiration (Fig. 5b). These areas typically have high topographic relief, combined with seasonal climate." We focused our analysis on the CONUS domain because we wanted to compare Prism and WorldClim as precipitation data sources, and fine resolution Prism data are only publicly available for CONUS. Thus while a global analysis would arguably be more comprehensive, it is not possible without acquiring (and paying for) proprietary Prism data.

The number of 1-degree by 1-degree grid cells (sampling points) at which heterogeneity biases are calculated per climate zones are now added to Figure 5b.

We present a table in the supplement in which we report statistical significance of differences between heterogeneity bias estimated at 1-degree by 1-degree grid cell across the contiguous US using 4 sets of P and PET data. The difference between heterogeneity bias estimated at the two climate zones that are raised by the reviewer (Cs and Ds) is not statistically significant across all 4 combinations of datasets (highlighted in yellow in Table S1 of the supplementary material). However, the difference between estimated heterogeneity bias in Cs vs Cf climate zones, and Ds versus Cf climate zones, as well as Cs versus Bs climate zones are significant across all four data combinations (highlighted in grey, blue, and green in Table S1 of the supplementary material). We discuss the main results of the statistical difference analysis in the manuscript (Section 4., second paragraph).

- The grid-scale dependence is tested for Switzerland and I don`t think we can "extrapolate" this exponential relationship everywhere around the globe. If you want to convince the reader you need to repeat this assessment for all regions (identified for instance in Fig. 3) where averaging effects are not negligible. Juxtaposing the different curves will (or will not) support the existance of a general "scaling" relationship with the grid resolution.

A global analysis would indeed be more comprehensive, but it would require high-resolution global data that we simply do not have. The graph of average heterogeneity bias versus grid resolution was added to figure 6 upon the reviewer's request in the previous round. In the manuscript, we report that "On average, the heterogeneity bias across Switzerland as a whole grows exponentially as the inputs are averaged over larger grids" and do not generalize it to any other region or the globe.

- In the first iteration I asked authors to provide more information about the value of "n" parameter. This information is still missing in the manuscript. Can the authors provide some concrete numbers on the sensitivity of their global estimates with respect to different "n" values?

This is an important point, thank you for raising it. We now present a sensitivity analysis for n values ranging from 2 to 5 in the supplement (Figures S1 and S2) and discuss its main results in the manuscript (Section 5. Summary and discussions, 5th paragraph). Those results are: 1) the spatial patterns of aggregation bias are similar, 2) the absolute magnitude of aggregation bias increases somewhat for higher values of n, as predicted by Eqs. 3 and 4, and 3) the Taylor approximation in Eqs. 3 and 4 yield realistic estimates of the aggregation bias for all values of n that were tested.

Other comments:
- Please do not include any discussion in the captions of the figures. See Figure 3-5-6.

We include concise statements of the main takeaway messages that the figures convey. We think that this is very helpful to readers – particularly those who scan the figures of a paper to get a first impression of its main points.  As this is a matter of style, we prefer to keep the captions as they are.

- Lines 36-37 in the abstract. I do not see how the results of this paper can be used for guiding a more detailed mechanistic modelling. Note also that averaging- or grid-scale effects have been largely reported also when using mechanistic models. So please remove this sentence.

**This sentence is removed from the abstract.**

- You do not want to quantify the absolute magnitude of the averaging effects and at the same you claim that your methodology is potentially a way for correcting such bias. The second statement imply a sort of quantification, in my opinion.

**This is correct, and we said this in the first paragraph of Section 5.  Obviously, to correct for aggregation biases one needs to quantify them.  As the first paragraph of Section 5 explains, the general approach outlined here could be used to quantify and correct for aggregation biases – but it would need to be applied to the mechanistic ET equations that are actually used in ESMs, rather than the simple Budyko curves that we have used here for purposes of illustration.**

- I found an imbalance between the emphasis you put on the introduction and the actual findings of the manuscript. As I said in the first iteration, this is an applied study of a previously described methodology that does not contain general insights.

**The entire introduction, except for the last paragraph, sets up the general problem of heterogeneity bias and how it is typically handled in ESMs.  This introduction is essential for readers who do not already know this material.  Although we do indeed apply a previously described methodology (and the introduction is quite explicit about this), the present paper presents a series of new insights, including:**

- **Our previous work showed mathematically that averaging over spatially heterogeneous P and PET results in overestimation of ET within the Budyko framework. We did not, however, determine where around the globe, and under what conditions, this heterogeneity bias is likely to be most important. In this work, we examine the global distribution of this bias, its scale dependence, and its sensitivity to variations in P versus PET.**

- **Our goal is to identify where, under what conditions, and at what spatial scales averaging over heterogeneities in P and PET could be most important to estimates of evapotranspiration, but not to quantify the absolute magnitude of these averaging effects.**

- **Our work outlines a strategy for quantifying heterogeneity biases and potentially correcting for them, and highlights regions where more detailed mechanistic modeling is needed.**

- **Our analysis of percent variability of P and PET products shows that percent variabilities of precipitation products are in general larger than PET products and hence contribute more to heterogeneity bias.**

- **Our analyses show that mountainous terrain, regions with temperate climates and dry summers, and landscapes where spatial variations in precipitation and potential evapotranspiration are inversely correlated exhibit greater heterogeneity bias in ET estimates.**

- **Our analysis of scale dependence (using Switzerland as a test case) shows that heterogeneity bias in Switzerland increases almost exponentially as gird cell sizes increase.**

- Lines 317-319 ("most likely due to the sharp spatial gradient…"). This is a quite generic statement.

**In this sentence and the next one, we are making exactly the statement that the reviewer said needed to be made (that the high aggregation biases in the Cs and Ds climatic zones are largely attributable to topography).**

**Hydrol. Earth Syst. Sci.**
**hess-2019-103**

**Dear Reviewer #3,**
**Thank you for your review and the detailed comments. Following please find our point by point response to your suggestions and questions. The Reviewer's comments are in regular font and our response is in bold.**

**Response to Referee #3 #report1**

Review of "Global assessment of how averaging over spatial heterogeneity in precipitation and potential evapotranspiration affects modeled evapotranspiration rates" by Elham Rouholahnejad Freund et al.

This is my first review of this work. The manuscript by Elham Rouholahnejad Freund et al. addresses the interesting issue of scaling of water and energy exchange at the land surface. While the manuscript focusses on a novel issue that could provide an interesting new addition to decades of literature on scaling, I do not find the manuscript in its current version to be convincing. This has to do with: a) a poor link between the main methodology and motivation as outlined in the Introduction, and b) a complete neglect of surface heterogeneity and scale-dependency in the Budyko n-parameter. In my view, the work could be a valuable contribution to HESS only if these deficiencies are addressed.

The work is motivated by potential scaling issues in ESMs. In my view, these relate mainly to effects of land surface heterogeneity (land use, soil type, groundwater tables, soil moisture, all of which can show large variability on small scales). To the degree these depend on forcing, this will to a large extent be caused by spatio-temporal variability of rainfall (i.e. convective storms leading to temporary wetting of part of a water-limited region only) and not just spatial variability. This is a big simplification where most of the scaling problem already is solved, and which is inherent to the choice for Budyko. A possible solution would be to solely focus on scaling issues within the Budyko framework due to P and PET. This would be a fairly novel approach, and it would avoid (artificially) linking too much to ESMs.

**Our analysis uses Budyko curves as a simple analytical framework (or a "see-through" function) to demonstrate our analysis. Our purpose is not to mimic the way that ESMs actually calculate ET (obviously so, since ESMs do not employ Budyko curves). Our purpose is instead to illustrate how variability in P and PET would be translated into biased ET estimates, using Budyko curves as a simple ET function for illustration purposes. Thus our purpose is not to highlight scaling issues within the Budyko framework per se.**

**We agree that Budyko curves already average over temporal heterogeneity (and the manuscript says this explicitly in the third paragraph of Section 3). It is unclear whether this temporal heterogeneity would lead to significant aggregation bias in ESMs, because they are usually solved on relatively short time steps.**

My second concern deals with the choice for a single Budyko n-parameter. Effectively, the authors show that at larger scales due to forcing heterogeneity, the Budyko curve tends to become more linear.

**This is indeed a consequence of our analysis, but it is not the point that we are trying to make. Our point is that _any_ nonlinear function will yield averages that lie "inside" the curve, and for ET functions this will always be below the curve. We are just using Budyko curves to illustrate this point.**

But what is the motivation for the baseline choice of n?

**For the calculations in the main paper, we used n=2 because this is a commonly used value in the existing literature. We will add this detail to the manuscript.**

Where is it shown that this value corresponds better to observations (i.e. is a more valid model) at finer scales (1 km) than at courser scales (1 degree)?

**We do not show this (and indeed we are not aware of any literature that does show it). This would require long-term catchment mass balances at the 1 km scale and at 1-degree scale, which are not widely available. In any case, our analysis is mainly concerned with the spatial pattern of aggregation bias (where is it larger? where is it smaller?) and this will not be particularly sensitive to the choice of n. We will add figures to the supplement where we compare aggregation bias calculations for different values of n.**

Would the results not strongly depend on the choice of n? In reality, n will also very strongly depend on land use (see for example Fig. 1 in https://www.hydrol-earth-syst-sci-discuss.net/hess-2018-634/#discussion or any of the many other studies on this subject). I think any analysis of scaling should focus on the main nonlinearities to avoid becoming a purely academic exercise (nothing wrong with the latter, but then it should be presented as such). As a minimum, I would expect a sensitivity analysis on how the results depend on the value of n, accompanied by a discussion on how n might vary locally and with scale.

**We present this sensitivity analysis (for n values ranging from 2 to 5) in the supplement (Figs. S1 and s2) and discuss its main results in the manuscript. Those results are: 1) the spatial patterns of aggregation bias are similar, 2) the absolute magnitude of aggregation bias increases somewhat for higher values of n, as predicted by Eqs. 3 and 4, and 3) the Taylor approximation in Eqs. 3 and 4 yield realistic estimates of the aggregation bias for all values of n that were tested.**

Ideally, I would see the manuscript being restructured towards a more theoretical analysis of effects of forcing heterogeneity on the Budyko model, which would result in a clearly testable hypothesis that Budyko curves should become more linear with increasing scale as a result of heterogeneity, and based on the maps regions can be identified where this effect should be largest and best observable. It should also be noted that the value of n used in the analysis is not reported, at least I could not find it.

**As explained above, our paper is not intended as an analysis of scaling effects in Budyko curves. We do not think that it is a particularly interesting hypothesis that Budyko curves should become more linear with increasing scale, since this is generally true of all curved functions (it is basically a mathematical theorem rather than an empirical hypothesis). We will report the value of n that we used, in addition to the results of the sensitivity analysis covering a range of n values.**

[revised manuscript text omitted]

---

## Author Response (AR3)

**Hydrol. Earth Syst. Sci.**
**hess-2019-103**

**Response to Reviewers comments – report #3**

**Dear Editor,**

**Following please find our point-by-point response to the two Referees' comments. Their comments are in regular font and our responses are in bold. The page and line numbers refer to the revised manuscript that will be submitted with this response (with "all mark up" display for review).**

**Referee #3, Report #2**

This is my second review of this manuscript. While I think it has improved considerably with respect to the previous version, in particular in the description and treatment of the Budyko-parameter, I remain somewhat skeptical about the generality of the results. I generally like scaling analysis, but I think one should always make a clear distinction between an analysis that is supposed to represent the real-world, or a hypothetical ("first-order"/potential/etc) analysis that shows the potential effect under certain restrictive assumptions (such as the use of a constant Budyko parameter in this case). The authors do acknowledge the limitations of their approach, but I feel this could (and should) be better reflected in the text. In my view, the manuscript would read much better if "heterogeneity bias" is replaced everywhere by "potential/theoretical/hypothetical heterogeneity bias" (or similar), so the suggestion of it being an analysis of the complete system including (varying) land surface properties is removed. I leave it up to the authors to make the relatively minor textual changes needed to accommodate this final comment.

**The term "heterogeneity bias" occurs 79 times in the text, and we do not think that adding "potential", "theoretical", or "hypothetical" 79 times would be an improvement. Where appropriate, we already refer to "estimated heterogeneity bias", "realistic estimates of the heterogeneity bias", "numerical estimates of the heterogeneity bias", "heterogeneity bias estimates", "heterogeneity bias in ET estimates", "heterogeneity bias in a hypothetical two-column model", and so forth. We also make clear that our goal is to identify spatial patterns in the heterogeneity bias, not its absolute magnitude.**

**We have reviewed all 79 occurrences of the phrase "heterogeneity bias", looking for any cases where it is not already clear from context that we are referring to a hypothetical calculation (or, alternatively, the general concept of heterogeneity bias, for which "hypothetical" would not apply anyhow). In the 12 cases where we could imagine that this was not already completely clear, we modified the text accordingly.**

**Referee #1, Report#3**

I highlight below few more points where, in my opinion, authors response is still not fully convincing. I think that an additional effort to improve the manuscript along these points could eventually make the work suitable for publication in HESS.

- In the first and second review iteration I made clear about the weak link between this work and ESMs. A pragmatic way of avoiding any misunderstanding on this point is to start the abstract (lines 22-25 of the track-changes version of the manuscript) and the first paragraph of the introduction (lines 41-48) in a different way. In any case, I would remove any link to ESMs that could potentially mislead the reader.

**We have removed any mention of ESM's from the abstract and have removed the first paragraph of the introduction almost entirely, line 42-49, (transferring only half a sentence into the second paragraph of the old introduction, which is now the first paragraph of the new introduction). These introductory statements now focus on "estimates of evapotranspiration" rather than ESM's.**

- I suggested extending the analysis of the heterogeneity bias by clustering the results over different climate zones over the globe. I think this analysis is still useful even if using one single dataset for P (i.e., WClim) and two datasets for PET (i.e., WClim and MODIS). This effort could lead to additional discussion that eventually elevate the scientific significance of the work.

**We appreciate the reviewer's perspective, but our own assessment of the usefulness of a global analysis is different. The advantages of confining this part of our analysis to the US are clear: we can compare the two precipitation data products (Prism and WorldClim) instead of having only one, and the observational constraints on both the P and PET data products are better in the US than over most of the globe. While we could of course ALSO do this analysis at global scale (but without Prism), we would prefer not to make the paper longer and more complicated at this stage.**

- I appreciate authors effort in providing additional insights on the implications of using different n values. However, I would expect some more explanations on the physical mechanisms leading to larger biases by higher n values. Finally, I do not think that repeating the analysis with spatially-distributed n values could create "artifacts". This additional analysis will show the interplay between scale-dependency in P, PET, and n with respect to the heterogeneity bias in ET. Again, this could be an effective way of bringing the scientific aspects of the work on the front.

**We now explain the effects of "n" as follows: "as expected from Eqs. 3 and 4, higher values of n lead to larger heterogeneity biases, because higher values of n localize the curvature of the Budyko function more strongly at the transition between the energy and water limits (Fig. 1b), increasing the heterogeneity bias for P/PET values near this transition." (Lines 393-400)**

**The problem with doing an analysis with spatially distributed n is this: how, and on what basis, should we assume that n varies from place to place? "n" is an empirical parameter, typically estimated by comparing mass balances from many catchments. Thus, it is an ensemble estimate over a group of catchments, and we do not have a solid basis for attributing different n values to individual catchments, let alone individual points on the landscape within catchments. Given that we have little or no information on how, and how much, n actually varies across real-world landscapes, we would prefer not to perform analyses based on arbitrary assumptions about spatial variability in n.**

[revised manuscript text omitted]